# Use of fluoroquinolones and the risk of aortic and mitral regurgitation: A nationwide case-crossover study

An-Hsun Chou[1], Chia-Pin Lin[2], Chun-Yu Chen[1], Victor Chien-Chia Wu[2], Yu-Ting Cheng[3], Yi-Hsin Chan[2], Fu-Chih Hsiao[2], Dong-Yi Chen[2], Kuo-Chun Hung[2], Pao-Hsien Chu[2], Shao-Wei Chen[3,4]*

1 Department of Anesthesiology, Chang Gung Memorial Hospital, Linkou Medical Center, Chang Gung University, Taoyuan City, Taiwan, 2 Department of Cardiology, Chang Gung Memorial Hospital, Linkou Medical Center, Chang Gung University, Taoyuan City, Taiwan, 3 Department of Surgery, Division of Thoracic and Cardiovascular Surgery, Chang Gung Memorial Hospital, Linkou Medical Center, Chang Gung University, Taoyuan City, Taiwan, 4 Center for Big Data Analytics and Statistics, Chang Gung Memorial Hospital, Linkou Medical Center, Taoyuan City, Taiwan

* josephchen0314@gmail.com

**Data Availability Statement:** The data underlying this study is from the National Health Insurance Research Database (NHIRD), which has been transferred to the Health and Welfare Data Science

## Abstract

### Background

Recently, there have been conflicting results reporting an increased risk of AR or MR associated with oral fluoroquinolones (FQs).This study investigated whether the use of FQs increases the risk of mitral regurgitation (MR) or aortic regurgitation (AR).

### Methods

A retrospective cohort study was conducted by using the Taiwan National Health Insurance research database. A unidirectional case-crossover design without selecting controls from an external population was adopted in this study. A total of 26,650 adult patients with new onset of AR or MR between January 1, 2000, and December 31, 2012, were identified. The risk of outcomes was compared between the hazard period and one of the randomly selected referent periods of the same individuals.

### Results

Before exclusion of pneumonia diagnosed within 2 months before the index date, patients who took FQs had a significantly greater risk of AR or MR (adjusted odds ratio [aOR] 1.51, 95% confidence interval [CI] 1.30–1.77), any AR (combined AR and MR) (aOR 1.50, 95% CI 1.10–2.04), and any MR (combined AR and MR) (aOR 1.37, 95% CI 1.16–1.62). After exclusion of pneumonia, FQs exposure remained significantly associated with a greater risk of MR (aOR 1.38, 95% CI 1.17–1.62) and any MR (aOR 1.25, 95% CI 1.05–1.48).

Center (HWDC). The NHIRD is not free to public access, and therefore interested researchers can obtain the data through formal application to the HWDC, Department of Statistics, Ministry of Health and Welfare, Taiwan (https://dep.mohw.gov.tw/DOS/cp-5119-59201-113.html). The authors had no special access privileges that others would not have.

**Funding:** This work was supported by a grant from Chang Gung Memorial Hospital, Taiwan (CFRPG3N0021, CMRPG3K1431, CMRPG3K1432, BMRPC19 (AHC), CORPG3M0371, CORPG3N0281, CFRPG3M0011, BMRPD95 (SWC)). This work was also supported by National Science and Technology Council grant NSTC-112-2314-B-182A-107 (SWC). The funders had no role in study design, data collection and analysis, decision to publish, or preparation of the manuscript.

**Competing interests:** The authors have declared that no competing interests exist.

## Conclusions

The findings suggested that patients treated with FQs could be warned about the potential risk for MR even after considering the possibility of protopathic bias. Reducing unnecessary FQs prescriptions may be considered to reduce the risk of valvular heart disease.

## Introduction

Fluoroquinolones (FQs) are broad-spectrum antibiotics with good oral bioavailability and are used for the treatment of a wide variety of infections [1,2]. FQs employ their antibacterial effect by preventing bacterial DNA from unwinding and duplicating, which takes place by the inhibition of bacterial topoisomerase and gyrase [3]. Studies have indicated that FQs cause degradation of collagen and extracellular matrix, potentially increasing the risk of tendinopathy, aortic aneurysms, and aortic dissections [4–7]. Furthermore, it has been suggested that FQ-induced collagen fiber degradation, especially the main component of type I collagen in native heart valves, could potentially affect the heart valves, leading to valvular regurgitation [8].

A recent study substantiated this claim with the finding of an increased risk of aortic and mitral valve regurgitation in patients treated with FQs [9]. Consequently, based on this experimental and observational study, the European Medicines Agency (EMA) published a safety concern against the use of FQs, especially in patient populations considered at high risk of regurgitant valvular heart disease (e.g., patients with hypertension or connective tissue disorders) [10]. However, in a recent nationwide nested case–control study, FQs were not significantly associated with increased rates of valvular regurgitation, and their results did not support a possible causal connection between FQs exposure and incident valvular regurgitation [11]. Additionally, a recent nationwide cohort study in Taiwan found no link between FQs and an increased risk of mitral or aortic regurgitation [12]. These conflicting results emphasize the need for further research to confirm or refute whether FQs increase the risk of aortic and mitral regurgitation. Therefore, we used a national database in Taiwan containing a population-based cohort with complete follow-up and detailed medication data to investigate the association of FQs with the risk of aortic and mitral regurgitation.

## Materials and methods

### Data source

This was designed as a self-controlled case-crossover study that utilized the Longitudinal Health Insurance Database 2000 (LHID2000). The LHID2000 is a subset of the National Health Insurance Research Database (NHIRD) and consists of claims data from 1 million randomly sampled individuals who were alive at the end of 2000. The coverage of LHID2000 was from 1997 to 2012. The LHID2000 has been validated to be representative of the general Taiwan population by the Taiwan National Health Research Institute, which maintains the NHIRD. Taiwan launched a National Health Insurance (NHI) program on March 1, 1995. The NHI system offers follow-up information on medications as well as on admission, outpatient clinic, and emergency department visit records of the Taiwanese population. The study received both ethical and administrative approval from the Institutional Review Board of Chang Gung Memorial Hospital (201901908B0, approved on 2019/12/27) and approval from the data holder, the NHI Administration. Informed consent for participants was waived.

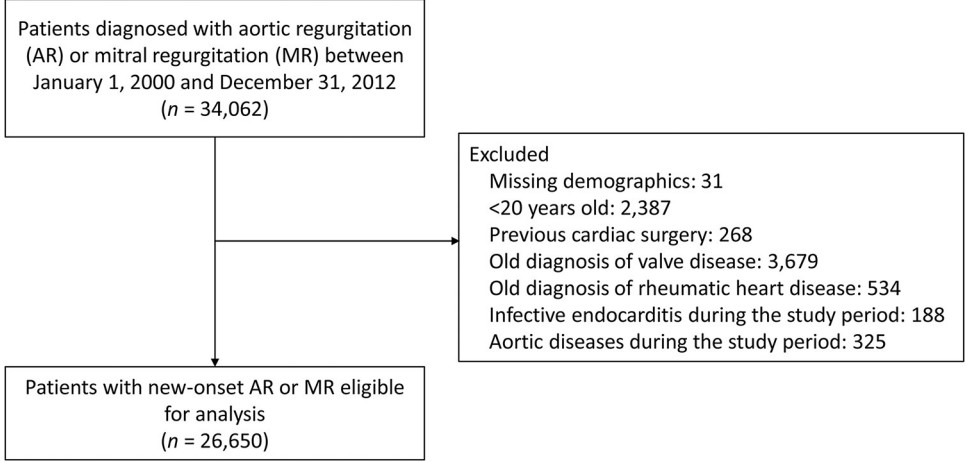

**Fig 1. Enrollment of the study patients.** *MR*, mitral regurgitation; *AR*, aortic regurgitation.

## Study population

Patients diagnosed with aortic regurgitation (AR) or mitral regurgitation (MR) between January 1, 2000, and December 31, 2012, were identified among the 1 million random samples of the entire Taiwanese population. The International Classification of Diseases–9th Revision-Clinical Modification (ICD-9-CM) diagnostic codes were used to identify patients with aortic valve or mitral valve disorders and aortic valve or mitral valve regurgitation (ICD-9: 424.0 and 424.1). At least three diagnoses in outpatient visits or any inpatient diagnosis were required to define AR or MR. We excluded those who were younger than 20 years (n = 2,387), those who previously underwent cardiac surgery (n = 268), those with an old diagnosis of any valve diseases (i.e., ICD-9: 394.9, 396.3, 396.8, 396.9, 424.0 and 424.1; n = 3,679), those with an old diagnosis of rheumatic heart disease (n = 534), those with infective endocarditis during the entire study period (n = 188), and those with aortic diseases during the entire study period (n = 325). The remaining 26,650 adult patients with new-onset AR or MR were eligible for analysis (**Fig 1**). The date of AR/MR diagnosis was assigned as the index date (also the day of outcome occurrence).

## Study design

By referring to a previous study that evaluated the association between the use of antibiotics and the risk of aortic diseases, a unidirectional case-crossover design without selecting controls from an external population was adopted in this study (**Fig 2**). One of the major advantages for the case-crossover design is the low threat of within-person time-invariant confounding (i.e., gene) and the ability to avoid selection biases from external controls. According to our design, each patient had one hazard period (-60 to -1 days), one washout period (-120 to -61 days), and three referent periods (-180 to -121 days, -240 to -181 days and -300 to -241 days) before the index date. The risk of outcomes was compared between the hazard period and one of the randomly selected referent periods of the same individuals. The antibiotic of primary interest were oral form drugs of fluoroquinolones (FQs). Other commonly prescribed antibiotics were selected as the negative control exposures, including amoxicillin, trimethoprim and sulfamethoxazole (TMP/SMX), amoxicillin clavulanate (AMC) and ampicillin sulbactam (ASB), and the second-generation drug cephalosporin. These antibiotics were chosen because their indications are generally similar to those of fluoroquinolones, according to treatment

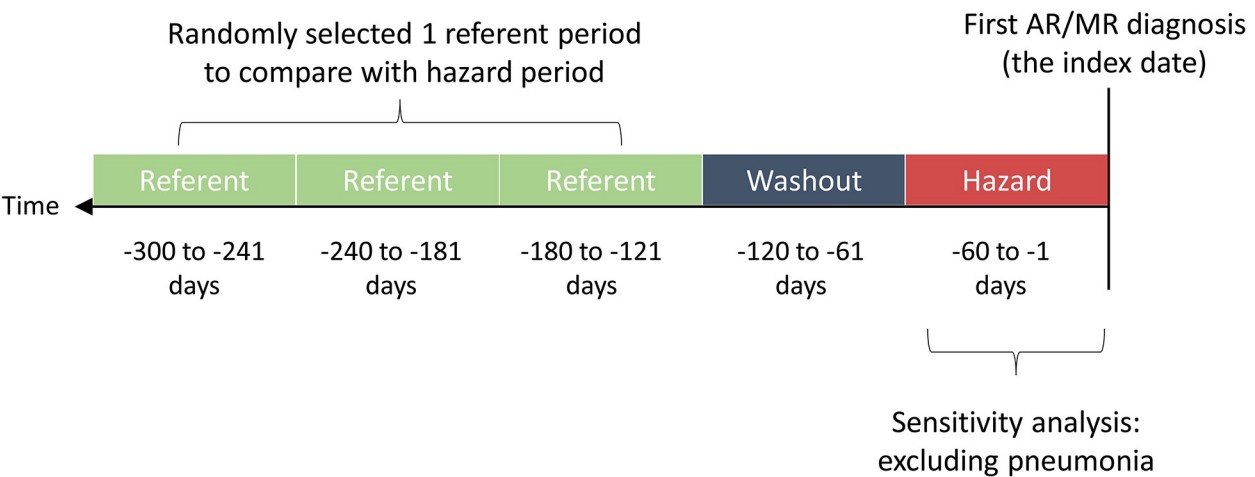

**Fig 2. Schematic representation of case-crossover studies.** *MR*, mitral regurgitation; *AR*, aortic regurgitation.

guidelines in Taiwan and other regions of Asia [12]. The definition of exposure was defined as a prescription ≥3 days in the outpatient department.

## Covariates

The covariates were age, sex, socioeconomic status (monthly income and urbanization level of the residence), and comorbidities (including old stroke, diabetes mellitus, hypertension, heart failure, coronary artery disease, atrial fibrillation, chronic kidney disease, dialysis, chronic obstructive pulmonary disease (COPD), liver cirrhosis and gastrointestinal bleeding history). The Charlson Comorbidity Index (CCI) score was also obtained. Comorbidities were defined as at least 2 outpatient diagnoses or 1 inpatient diagnosis in the previous year. The medications were extracted from the claims data of outpatient departments or refilling prescriptions for chronic illness in the previous six months. The medications considered to be relevant to this study were angiotensin-converting enzyme inhibitors (ACEIs), angiotensin receptor blockers (ARBs), beta-blockers, dihydropyridine calcium-channel blockers (dCCBs), diuretics, anti-platelet drugs, anticoagulants, statins, digoxin, oral diabetes drugs, and insulin.

## Statistical analysis

The baseline demographic and clinical characteristics of patients with different diagnoses (MR alone, AR alone and both AR and MR) were compared using one-way analysis of variance for continuous variables and the chi-square test for categorical variables. The risk of outcomes (AR or MR, any AR and any MR) for one patient between the hazard period and the randomly selected referent period was compared using a conditional logistic regression model. The use of other medications may change with time for one patient and could be related to outcomes; therefore, the time-varying use of other medications by each month was adjusted in the multi-variable model. The clinical representations and symptoms are similar between pneumonia and AR/MR, so antibiotics in the hazard period may be prescribed for treating pneumonia, not for treating AR/MR. This phenomenon is known as "protopathic bias", which would induce reverse causality. Therefore, the aforementioned analyses were performed by excluding patients who were diagnosed with pneumonia (during either inpatient or outpatient visits) in the hazard period. It should be noted that the primary analysis was based on the results after excluding pneumonia during the hazard period in the study. A two-sided *P* value <0.05 was

considered statistically significant. Statistical analyses were performed using SAS version 9.4 (SAS Institute, Cary, NC).

## Results

### Clinical characteristics of the patients

After applying the exclusion criteria, a total of 26,650 patients were eligible for inclusion in this cohort (Fig 1). The mean age was 54.1 years (standard deviation [SD] = 18.6 years), and males were less predominant (38%). Prevalent comorbidities were hypertension (35.9%) and coronary artery disease (20.8%) (Table 1). Subsequently, the cohort was separated into subcohorts of patients with MR alone, AR alone or combined AR and MR. We identified a total of 20,884 cases of MR, 3,940 cases of AR and 1,866 cases of combined AR and MR. Among the three cohorts, the mean age was much younger in MR than in AR and combined AR and MR (49.9 vs. 70.9 vs. 65.9 years). Males were more predominant in AR than in MR and combined AR and MR (49.6% vs. 35.4% vs. 42%). The most common comorbidity in the three cohorts was hypertension. Among them, hypertension was dominated by AR (65.9%), followed by combined AR and MR (52.1%) and MR (28.8%). The CCI score was higher in AR than in MR and combined AR and MR (1.6 vs. 0.8 vs. 1.1). Beta-blockers, dCCB and ACEi/ARB were the most commonly used medications in the three cohorts.

### Association of the antibiotics use and the risk of AR/MR

Table 2 estimates the relationship between exposure to antibiotics and the risk of outcomes (AR or MR, any AR and any MR). After adjusting for the time-varying use of other medications, the results demonstrated that patients who took FQs (adjusted odds ratio [aOR] 1.51, 95% confidence interval [CI] 1.30–1.77), TMP/SMX (aOR 1.20, 95% CI 1.04–1.39), and AMC/ASB (aOR 1.74, 95% CI 1.39–2.16) had a significantly greater risk of AR or MR in the hazard period than in the reference period. In terms of the risk of any AR (including combined AR and MR), exposures to FQs (aOR 1.50, 95% CI 1.10–2.04), AMC/ASB (aOR 2.58, 95% CI 1.62–4.11) and second-generation cephalosporin (aOR 2.51, 95% CI 1.38–4.55) showed significantly positive associations. Concerning the risk of any MR (including combined AR and MR), exposures to FQs (aOR 1.37, 95% CI 1.16–1.62), amoxicillin (aOR 1.10, 95% CI 1.01–1.21) and AMC/ASB (aOR 1.63, 95% CI 1.27–2.10) demonstrated significantly positive associations.

### Association of the antibiotics use and the risk of AR/MR after excluding patients diagnosed with pneumonia during the hazard period

Subsequently, in our study, we excluded those who had pneumonia diagnosed within 2 months before the index date (the hazard period) and then reanalyzed the relationship between exposure to antibiotics and the risk of outcomes (Table 3). There were 500 patients with a diagnosis of pneumonia in the hazard period. Obliviously, the strength of the associations became weaker, reflected by the smaller odds ratios (closer to 1) and the lower significance (larger *P* values). The results revealed that exposure to FQs was still significantly associated with a greater risk of AR or MR (aOR 1.38, 95% CI 1.17–1.62) and any MR (aOR 1.25, 95% CI 1.05–1.48). However, exposure to FQs was not significantly correlated with a larger risk of any AR (*P* = 0.101). In addition, the use of TMP/SMX was also significantly associated with a higher risk of AR or MR (aOR 1.18, 95% CI 1.02–1.37).

**Table 1. Demographic and clinical characteristics of patients at baseline.**

| Variable | Total (N = 26,650) | MR alone (n = 20,884) | AR alone (n = 3,940) | AR+MR (n = 1,866) | P value |
|---|---|---|---|---|---|
| Age (years) | 54.1 ± 18.6 | 49.9 ± 17.6 | 70.9 ± 12.9 | 65.9 ± 13.9 | <0.001 |
| Male sex | 10,120 (38.0) | 7,381 (35.4) | 1,955 (49.6) | 784 (42.0) | <0.001 |
| Monthly income | | | | | <0.001 |
| Tertile 1 | 8,224 (30.9) | 13,463 (64.6) | 1,985 (50.4) | 1,082 (58.0) | |
| Tertile 2 | 9,448 (35.5) | 5,632 (27.0) | 1,798 (45.6) | 794 (42.6) | |
| Tertile 3 | 8,978 (33.7) | 7,124 (34.2) | 1,598 (40.6) | 726 (38.9) | |
| Urbanization level | | | | | <0.001 |
| Low | 2,975 (11.2) | 2,098 (10.1) | 608 (15.4) | 269 (14.4) | |
| Moderate | 7,464 (28.0) | 5,469 (26.2) | 1,370 (34.8) | 625 (33.5) | |
| High | 7,636 (28.7) | 6,161 (29.6) | 1,002 (25.4) | 473 (25.3) | |
| Very High | 8,575 (32.2) | 7,116 (34.1) | 960 (24.4) | 499 (26.7) | |
| Comorbid conditions | | | | | |
| Old stroke | 1,327 (5.0) | 807 (3.9) | 407 (10.3) | 113 (6.1) | <0.001 |
| Diabetes mellitus | 2,936 (11.0) | 1,967 (9.4) | 743 (18.9) | 226 (12.1) | <0.001 |
| Hypertension | 9,571 (35.9) | 6,002 (28.8) | 2,596 (65.9) | 973 (52.1) | <0.001 |
| Heart failure | 1,044 (3.9) | 707 (3.4) | 263 (6.7) | 74 (4.0) | <0.001 |
| Coronary artery disease | 5,537 (20.8) | 3,731 (17.9) | 1,303 (33.1) | 503 (27.0) | <0.001 |
| Atrial fibrillation | 1,118 (4.2) | 809 (3.9) | 225 (5.7) | 84 (4.5) | <0.001 |
| Chronic kidney disease | 1,688 (6.3) | 1,082 (5.2) | 465 (11.8) | 141 (7.6) | <0.001 |
| Dialysis | 259 (1.0) | 167 (0.8) | 67 (1.7) | 25 (1.3) | <0.001 |
| COPD | 2,053 (7.7) | 1,217 (5.8) | 621 (15.8) | 215 (11.5) | <0.001 |
| Liver cirrhosis | 238 (0.89) | 162 (0.78) | 62 (1.57) | 14 (0.75) | <0.001 |
| Gastrointestinal bleeding history | 2,436 (9.1) | 1,587 (7.6) | 638 (16.2) | 211 (11.3) | <0.001 |
| Charlson's Comorbidity Index score | 0.9 ± 1.5 | 0.8 ± 1.4 | 1.6 ± 1.8 | 1.1 ± 1.5 | <0.001 |
| Medication at the index date | | | | | |
| ACEi/ARB | 7,115 (26.7) | 4,463 (21.4) | 1,899 (48.2) | 753 (40.4) | <0.001 |
| Beta-blocker | 11,035 (41.4) | 8,228 (39.5) | 1,921 (48.8) | 886 (47.5) | <0.001 |
| dCCB | 7,272 (27.3) | 4,465 (21.4) | 2,056 (52.2) | 751 (40.2) | <0.001 |
| Diuretics | 3,365 (12.6) | 2,126 (10.2) | 896 (22.7) | 343 (18.4) | <0.001 |
| Antiplatelets | 6,314 (23.7) | 4,132 (19.8) | 1,559 (39.6) | 623 (33.4) | <0.001 |
| Anticoagulation | 363 (1.4) | 262 (1.3) | 69 (1.8) | 32 (1.7) | 0.019 |
| Statin | 2,648 (9.9) | 1,868 (9.0) | 573 (14.5) | 207 (11.1) | <0.001 |
| Digoxin | 1,422 (5.3) | 917 (4.4) | 330 (8.4) | 175 (9.4) | <0.001 |
| Oral diabetes drugs | 2,503 (9.4) | 1,650 (7.9) | 645 (16.4) | 208 (11.1) | <0.001 |
| Insulin | 606 (2.3) | 406 (1.9) | 168 (4.3) | 32 (1.7) | <0.001 |

Abbreviations: *MR*, mitral regurgitation; *AR*, aortic regurgitation; *COPD*, chronic obstructive pulmonary disease; *ACEi*, angiotensin converting enzyme inhibitor; *ARB*, angiotensin II receptor blocker; *dCCB*, dihydropyridine calcium channel blocker.

Data were presented as frequency and percentage or mean ± standard deviation.

## Discussion

In this nationwide self-controlled case-crossover study, before excluding pneumonia diagnosed within 2 months before the index date, the results showed an association between FQs and AR or MR, any AR and any MR. TMP/SMX and AMC/ASB for AR or MR, AMC/ASB and cephalosporin for any AR, and AMC and AMC/ASB for any MR also had greater risk similar to FQs. After excluding pneumonia, although FQs were not significantly correlated with a

**Table 2. Association of the antibiotics use and the risk of AR/MR.**

| Outcome / drug | No. of prescription / No. of patient (%) | | Unadjusted analysis | | Adjusted analysis* | |
|---|---|---|---|---|---|---|
| | Hazard period | Referent period | OR (95% CI) | P value | OR (95% CI) | P value |
| AR or MR | | | | | | |
| Fluoroquinolone | 491 / 26,650 (1.8%) | 344 / 26,650 (1.3%) | 1.50 (1.29–1.74) | <0.001 | 1.51 (1.30–1.77) | <0.001 |
| Amoxicillin | 1,502 / 26,650 (5.6%) | 1,413 / 26,650 (5.3%) | 1.07 (0.99–1.16) | 0.075 | 1.05 (0.97–1.14) | 0.216 |
| TMP/ SMX | 505 / 26,650 (1.9%) | 437 / 26,650 (1.6%) | 1.18 (1.03–1.35) | 0.019 | 1.20 (1.04–1.39) | 0.012 |
| AMC/ ASB | 271 / 26,650 (1.02%) | 145 / 26,650 (0.54%) | 1.97 (1.60–2.43) | <0.001 | 1.74 (1.39–2.16) | <0.001 |
| Cephalosporin 2nd | 145 / 26,650 (0.54%) | 113 / 26,650 (0.42%) | 1.31 (1.01–1.68) | 0.040 | 1.19 (0.91–1.55) | 0.208 |
| AR (including AR+MR) | | | | | | |
| Fluoroquinolone | 131 / 5,806 (2.3%) | 86 / 5,806 (1.5%) | 1.65 (1.23–2.23) | 0.001 | 1.50 (1.10–2.04) | 0.012 |
| Amoxicillin | 359 / 5,806 (6.2%) | 305 / 5,806 (5.3%) | 1.21 (1.03–1.42) | 0.024 | 1.17 (0.99–1.39) | 0.068 |
| TMP/ SMX | 110 / 5,806 (1.9%) | 94 / 5,806 (1.6%) | 1.21 (0.89–1.63) | 0.221 | 1.13 (0.82–1.54) | 0.462 |
| AMC/ ASB | 82 / 5,806 (1.41%) | 30 / 5,806 (0.52%) | 3.08 (1.96–4.84) | <0.001 | 2.58 (1.62–4.11) | <0.001 |
| Cephalosporin 2nd | 42 / 5,806 (0.72%) | 18 / 5,806 (0.31%) | 2.50 (1.40–4.46) | 0.002 | 2.51 (1.38–4.55) | 0.003 |
| MR (including AR+MR) | | | | | | |
| Fluoroquinolone | 401 / 22,750 (1.8%) | 303 / 22,750 (1.3%) | 1.37 (1.17–1.60) | <0.001 | 1.37 (1.16–1.62) | <0.001 |
| Amoxicillin | 1,263 / 22,750 (5.6%) | 1,149 / 22,750 (5.1%) | 1.12 (1.02–1.22) | 0.013 | 1.10 (1.01–1.21) | 0.032 |
| TMP/ SMX | 432 / 22,750 (1.9%) | 392 / 22,750 (1.7%) | 1.12 (0.97–1.29) | 0.136 | 1.13 (0.97–1.32) | 0.116 |
| AMC/ ASB | 201 / 22,750 (0.89%) | 114 / 22,750 (0.50%) | 1.83 (1.44–2.32) | <0.001 | 1.63 (1.27–2.10) | <0.001 |
| Cephalosporin 2nd | 107 / 22710 (0.47%) | 100 / 22,750 (0.44%) | 1.07 (0.81–1.42) | 0.618 | 1.00 (0.75–1.34) | 0.994 |

Abbreviations: *MR*, mitral regurgitation; *AR*, aortic regurgitation; *OR*, odds ratio; *CI*, confidence interval; *TMP*, trimethoprim; *SMX*, sulfamethoxazole; *AMC*,
amoxicillin clavulanate; *ASB*, ampicillin sulbactam.

*Adjusted for time-varying use of medications other than antibiotics.

greater risk of any AR, FQs were still significantly associated with greater risks of AR or MR and any MR.

Recently, most studies have identified which fluoroquinolones appear to upregulate multiple matrix metalloproteinases, leading to collagen fibril degradation and an increased risk of collagen-related adverse events [4–7]. In the aortic wall, type I and type III are the dominant forms of collagen, thereby suggesting that FQs lead to aortic aneurysms and dissection [6,7]. There is also some evidence that FQ exposure is related to heart valve prolapse. In a case report where a patient who took ciprofloxacin 750 mg twice daily for 2 days, aortic valve prolapse occurred [13]. A recent study that used the FDA's adverse reporting system database revealed an approximately 2-fold increased risk of MR and AR with current or previous use of oral FQs compared with patients taking amoxicillin or azithromycin [9]. Additionally, an experimental study reported that exposure to ciprofloxacin led to collagen degradation in aortic myofibroblast cells isolated from patients with aortopathy, including AR (8). With these findings, EMA has informed health care professionals of the risk of heart valve regurgitation/insufficiency associated with systemic and inhaled use of FQs [10]. However, a Danish nationwide nested case–control study refuted this claim. They reported the lack of an association between FQs exposure in patients with AR or MR, either in high-risk patients with hypertension or when investigating potential dose–response effects [11]. These results are somewhat contradictory. The difference in prescription patterns between antibiotics and different comparator drugs in these studies may be the reason for the difference in results. These conflicting results underline the need for further large-scale studies to investigate the association of FQs with the risk of AR and MR.

In our nationwide case-crossover study, we use a unidirectional case-crossover design, the main advantage of which is the low threat of within-person time-invariant confounding and

**Table 3. Association of the antibiotics use and the risk of AR/MR, excluding patients diagnosed with pneumonia during the hazard period.**

| Outcome / drug | No. of prescription / No. of patient (%) | | Unadjusted analysis | | Adjusted analysis* | |
|---|---|---|---|---|---|---|
| | Hazard period | Referent period | OR (95% CI) | P value | OR (95% CI) | P value |
| AR or MR | | | | | | |
| Fluoroquinolone | 429 / 26,150 (1.6%) | 327 / 26,150 (1.3%) | 1.36 (1.17–1.59) | <0.001 | 1.38 (1.17–1.62) | <0.001 |
| Amoxicillin | 1,456 / 26,150 (5.6%) | 1,390 / 26,150 (5.3%) | 1.06 (0.98–1.14) | 0.181 | 1.04 (0.96–1.13) | 0.389 |
| TMP/ SMX | 485 / 26,150 (1.9%) | 426 / 26,150 (1.6%) | 1.16 (1.01–1.33) | 0.038 | 1.18 (1.02–1.37) | 0.022 |
| AMC/ ASB | 181 / 26,150 (0.69%) | 137 / 26,150 (0.52%) | 1.34 (1.07–1.69) | 0.011 | 1.22 (0.96–1.54) | 0.110 |
| Fluoroquinolone | 119 / 26,150 (0.46%) | 109 / 26,150 (0.42%) | 1.10 (0.84–1.44) | 0.494 | 1.03 (0.78–1.37) | 0.815 |
| AR (including AR+MR) | | | | | | |
| Fluoroquinolone | 112 / 5,626 (2.0%) | 83 / 5,626 (1.5%) | 1.44 (1.05–1.97) | 0.023 | 1.32 (0.95–1.83) | 0.101 |
| Amoxicillin | 341 / 5,626 (6.1%) | 296 / 5,626 (5.3%) | 1.18 (0.996–1.39) | 0.055 | 1.14 (0.96–1.36) | 0.139 |
| TMP/ SMX | 104 / 5,626 (1.8%) | 90 / 5,626 (1.6%) | 1.19 (0.87–1.63) | 0.269 | 1.10 (0.80–1.52) | 0.573 |
| AMC/ ASB | 43 / 5,626 (0.76%) | 27 / 5,626 (0.48%) | 1.70 (1.01–2.84) | 0.045 | 1.53 (0.90–2.60) | 0.118 |
| Cephalosporin 2nd | 31 / 5,626 (0.55%) | 18 / 5,626 (0.32%) | 1.81 (0.98–3.34) | 0.056 | 1.85 (0.99–3.47) | 0.055 |
| MR (including AR+MR) | | | | | | |
| Fluoroquinolone | 355 / 22,356 (1.6%) | 294 / 22,356 (1.3%) | 1.24 (1.05–1.46) | 0.011 | 1.25 (1.05–1.48) | 0.013 |
| Amoxicillin | 1,229 / 22,356 (5.5%) | 1,132 / 22,356 (5.1%) | 1.10 (1.01–1.20) | 0.032 | 1.09 (0.99–1.19) | 0.076 |
| TMP/ SMX | 417 / 22,356 (1.9%) | 380 / 22,356 (1.7%) | 1.11 (0.96–1.29) | 0.163 | 1.13 (0.97–1.32) | 0.116 |
| AMC/ ASB | 142 / 22,356 (0.64%) | 109 / 22,356 (0.49%) | 1.32 (1.02–1.71) | 0.033 | 1.20 (0.91–1.57) | 0.193 |
| Cephalosporin 2nd | 90 / 22,356 (0.40%) | 100 / 22,356 (0.45%) | 0.90 (0.67–1.20) | 0.456 | 0.84 (0.62–1.15) | 0.279 |

Abbreviations: *MR*, mitral regurgitation; *AR*, aortic regurgitation; *OR*, odds ratio; *CI*, confidence interval; *TMP*, trimethoprim; *SMX*, sulfamethoxazole; *AMC*, amoxicillin clavulanate; *ASB*, ampicillin sulbactam.

*Adjusted for time-varying use of medications other than antibiotics.

avoidance of selection bias from external controls. Moreover, we used several oral antibiotics suggested by the treatment guidelines in Taiwan, including FQs, AMC, TMP/SMX, AMC/ASB, and the second-generation cephalosporin. In this study, the results showed associations between FQs and AR or MR, any AR and any MR. TMP/SMX and AMC/ASB for AR or MR, AMC/ASB and cephalosporin for any AR, and AMC and AMC/ASB for any MR also had greater risks similar to FQs. Our results showed that most antibiotics were associated with valvular heart disease. We believe this may be due to treatment of a certain infection with symptoms similar to valvular insufficiency. Clinically, the symptoms of pneumonia and valvular insufficiency are difficult to clearly differentiate before echocardiographic evaluation. The clinical signs and symptoms frequently overlap between heart failure due to heart valve regurgitation and pneumonia [14]. Exertional breathlessness, nocturnal cough, and paroxysmal nocturnal dyspnea are common to both conditions [15]. CXR is also difficult to differentiate between them [16]. It is unable to differentiate that the empiric antibiotic treatment before echocardiographic diagnosis is due to the symptoms caused by pneumonia or heart failure due to valvular insufficiency [17]. Antibiotics may be prescribed for treating pneumonia rather than valvular insufficiency. To further strengthen our results, we excluded those who had pneumonia diagnosed within 2 months before the index date. After excluding pneumonia, we still found an association between FQ use and the greater risk of AR or MR and any MR, although any AR was not associated with exposure to FQs. We agree that the use of FQs is related to the risk of MR, and FQs may have a probable or possible causal association with AR. Future studies are necessary to confirm these associations. FQs should only be used after carefully assessing their likely benefits and risks, including those of AR or MR.

A recent nationwide cohort study in Taiwan found no link between FQs and an increased risk of mitral or aortic regurgitation [12]. They employed an active comparative study design, comparing FQs with amoxicillin/clavulanate among different individuals. In contrast, our study indicates an increased risk of AR or MR in patients treated with FQs. Several key methodological differences may account for these contrasting findings. Our study utilized a unidirectional case-crossover design, comparing hazard periods and referent periods within the same individuals. This design helps to minimize within-person time-invariant confounding factors such as genetic predispositions, thereby reducing potential biases associated with external control groups. Conversely, the previous study's active comparative design, comparing FQs with amoxicillin/clavulanate among different individuals, can introduce various confounding factors and biases that are not present in a case-crossover design. Additionally, their study differentiated between respiratory and non-respiratory FQs, which could further complicate the interpretation of their results. Despite these conflicting results, it is crucial to recognize the inherent limitations and differences in study designs when interpreting these findings. Further research is warranted to confirm or refute the association between FQs and the risk of aortic and mitral regurgitation, taking into account the various confounding factors and biases.

The use of a population-based dataset comprising a large number of subjects and the unidirectional case-crossover design could be considered the strengths of this study. However, limitations are still present in this study. First, when using nationwide health care registers and ICD-9-CM codes, there is the potential for misclassification of diagnostic and procedural code errors. However, the accuracy of the diagnosis and procedural codes for the major cardiovascular diseases have been validated in studies, and NHIRD has a high accuracy in diagnosis and is a valid source for research in cardiovascular diseases [18,19]. Second, misclassification of FQs exposure is possible, and we cannot be sure that patients complied with treatment. We assumed accurate correspondence between refilling records and drug intake, but we cannot exclude the possibility of misclassification of exposure. However, such exposure misclassification would be random, and any effect on our results would thus be limited. Furthermore, some important clinical information, such as blood pressure, imaging findings, echocardiographic diagnosis of valvular lesions, microbiology testing results, and laboratory data, were sometimes lacking, which may have led to confounders, such as infection severity. We could only investigate associations without inferring causation, and the possibility of protopathic bias could not be completely excluded. Despite these limitations, we believe our results provide a valuable contribution to the knowledge gap in this field.

## Conclusion

Before the exclusion of pneumonia, most antibiotics were associated with valvular insufficiency. After the exclusion of pneumonia, there were still associations between FQs use and the greater risk of MR and any MR. Our results suggest that the association of other antibiotics with valvular heart disease should be due to the interference of pneumonia. The findings suggested that patients treated with FQs should be warned about the potential risk for MR. Reducing unnecessary FQs prescriptions may be considered to reduce the risk of valvular heart disease.

## Acknowledgments

This study was based on data from the NHIRD provided by the NHI administration, Ministry of Health and Welfare of Taiwan, and managed by the National Health Research Institutes of Taiwan. However, the interpretation and conclusions contained in this paper only represent

the authors. The authors thank the statistical assistance and wish to acknowledge the support of the Maintenance Project of the Center for Big Data Analytics and Statistics, Chang Gung Memorial Hospital, Linkou for study design and monitor, data analysis and interpretation. The authors also thank Alfred Hsing-Fen Lin and Zoe Ya-Jhu Syu for their assistance with the statistical analysis.

## Author Contributions

**Conceptualization:** An-Hsun Chou, Chia-Pin Lin, Pao-Hsien Chu, Shao-Wei Chen.

**Data curation:** An-Hsun Chou, Chun-Yu Chen, Pao-Hsien Chu, Shao-Wei Chen.

**Formal analysis:** An-Hsun Chou, Chia-Pin Lin, Victor Chien-Chia Wu, Pao-Hsien Chu, Shao-Wei Chen.

**Funding acquisition:** An-Hsun Chou, Shao-Wei Chen.

**Investigation:** An-Hsun Chou, Chia-Pin Lin, Chun-Yu Chen, Yu-Ting Cheng, Yi-Hsin Chan.

**Methodology:** Victor Chien-Chia Wu, Yu-Ting Cheng, Yi-Hsin Chan.

**Project administration:** Chun-Yu Chen, Fu-Chih Hsiao.

**Resources:** An-Hsun Chou, Fu-Chih Hsiao, Dong-Yi Chen.

**Software:** Yu-Ting Cheng, Yi-Hsin Chan, Fu-Chih Hsiao, Dong-Yi Chen, Kuo-Chun Hung.

**Supervision:** Chia-Pin Lin, Victor Chien-Chia Wu, Yu-Ting Cheng, Yi-Hsin Chan, Fu-Chih Hsiao, Dong-Yi Chen, Kuo-Chun Hung, Shao-Wei Chen.

**Validation:** An-Hsun Chou, Chun-Yu Chen, Kuo-Chun Hung, Pao-Hsien Chu, Shao-Wei Chen.

**Visualization:** An-Hsun Chou, Chia-Pin Lin, Kuo-Chun Hung, Pao-Hsien Chu.

**Writing – original draft:** An-Hsun Chou, Shao-Wei Chen.

**Writing – review & editing:** Shao-Wei Chen.

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
