## [Decision Letter · Decision Letter 0]

13 May 2024

PONE-D-24-09908Use of fluoroquinolones and the risk of aortic and mitral regurgitation: a nationwide case-crossover studyPLOS ONE

Dear Dr. Chen,

Thank you for submitting your manuscript to PLOS ONE. After careful consideration, we feel that it has merit but does not fully meet PLOS ONE’s publication criteria as it currently stands. Therefore, we invite you to submit a revised version of the manuscript that addresses the points raised during the review process.

We look forward to receiving your revised manuscript.

Kind regards,

Yoshihiro Fukumoto

Academic Editor

PLOS ONE

https://journals.plos.org/plosone/s/file?id=ba62/PLOSOne_formatting_sample_title_authors_affiliations.pdf"

2.Thank you for stating the following financial disclosure: 

"This work was supported by a grant from Chang Gung Memorial Hospital, Taiwan (CFRPG3N0021, CMRPG3K1431, CMRPG3K1432, BMRPC19 (AHC), CORPG3M0371, CORPG3N0281, CFRPG3M0011, BMRPD95 

 (SWC)). This work was also supported by National Science and Technology Council grant  NSTC-112-2314-B-182A-107 (SWC)."

3. We note that your Data Availability Statement is currently as follows: [All relevant data are within the manuscript and its Supporting Information files]

Reviewers' comments:

Reviewer's Responses to Questions

**Comments to the Author**

1. Is the manuscript technically sound, and do the data support the conclusions?

Reviewer #1: Yes

Reviewer #2: Yes

2. Has the statistical analysis been performed appropriately and rigorously? 

Reviewer #1: Yes

Reviewer #2: Yes

3. Have the authors made all data underlying the findings in their manuscript fully available?

Reviewer #1: No

Reviewer #2: Yes

4. Is the manuscript presented in an intelligible fashion and written in standard English?

Reviewer #1: Yes

Reviewer #2: Yes

5. Review Comments to the Author

Reviewer #1: This is an interesting study to evaluate the association between FQs prescriptions and the risk of valvular heart diseases. I have a few comments to the authors.

1. Novelity: There is a Meta-Analysis in Drug Saf. 2019 Apr;42(4):529-538. "Fluoroquinolones and Cardiovascular Risk: A Systematic Review, Meta-analysis and Network Meta-analysis" PMID: 30368737 What is the novelty of this study?

2. The authors highlight interference of pneumonia. Why they focus on pnumonia? There are other infectious disease. Especially, infective endocarditis is a major cause for valvular regurgetation.

3. The authors also observed the association between other antibiotics and valuvlar insufficency. What is the plausibility of this observation?

Reviewer #2: This was a very interesting study that explored the relationship between FQs and the onset of valvular heart disease.However, I have some questions about this paper.

1.Rather than suggesting that exposure to FQs poses a risk for MR and AR, the results presented in this paper give the impression that antibiotics are associated with the development of valvular heart disease.

2.Although the purpose of thi study was to explore the risks of FQs, it is not clear why amoxicillin, TMP/SMX, AMC and ASB, and second-generation cephalosporin drugs were selected in addition to FQs.

3.A national cohort study in Taiwan did not show any association between FQs and the onset of valvular heart disease. Please comment on the differences with this previous study.

Clin Pharmacol Ther. 2024 Jan;115(1):147-157. doi: 10.1002/cpt.3084. Epub 2023 Nov 13.

6. PLOS authors have the option to publish the peer review history of their article (what does this mean?). If published, this will include your full peer review and any attached files.

Reviewer #1: No

Reviewer #2: No

---

## [Author Response · Author response to Decision Letter 0]

20 Jun 2024

Dear Dr. Yoshihiro Fukumoto, 

Academic Editor

PLOS ONE

PONE-D-24-09908

"Use of fluoroquinolones and the risk of aortic and mitral regurgitation: a nationwide case-crossover study"

Thank you for providing the additional requirements for our manuscript revision. We appreciate the guidance and will ensure that our submission meets PLOS ONE's standards.

On behalf of the authors, I would like to express our gratitude for your detailed feedback. We have carefully reviewed your comments and revised the manuscript accordingly. 

We appreciate your attention to these details and look forward to submitting our revised manuscript. Please let us know if there are any further requirements or clarifications needed.

Best regards,

Shao-Wei Chen

 

Reviewer #1: This is an interesting study to evaluate the association between FQs prescriptions and the risk of valvular heart diseases. I have a few comments to the authors.

1. Novelity: There is a Meta-Analysis in Drug Saf. 2019 Apr;42(4):529-538. "Fluoroquinolones and Cardiovascular Risk: A Systematic Review, Meta-analysis and Network Meta-analysis" PMID: 30368737 What is the novelty of this study?

Response: Thank you for your comment. The referenced meta-analysis primarily addresses the risks of arrhythmia, torsades de pointes, and cardiovascular mortality associated with fluoroquinolones. Additionally, recent studies have highlighted the association between fluoroquinolone use and an increased risk of aortic dissection and aneurysm.

Our study extends the understanding of fluoroquinolone-associated cardiovascular risks by focusing specifically on mitral and aortic regurgitation. We employ a unique methodology and utilize a distinctive data source, providing new insights into this area. These contributions are significant and novel, complementing and expanding upon the findings of previous studies.

2. The authors highlight interference of pneumonia. Why they focus on pnumonia? There are other infectious disease. Especially, infective endocarditis is a major cause for valvular regurgetation.

Response: Thank you for your comment. We focused on pneumonia because its symptoms can closely resemble those of valvular regurgitation, potentially confounding the diagnosis and analysis of AR/MR. This concern was thoroughly detailed in our methodology section to ensure clarity as “The clinical representations and symptoms are similar between pneumonia and AR/MR, so antibiotics in the hazard period may be prescribed for treating pneumonia, not for treating AR/MR. This phenomenon is known as “protopathic bias”, which would induce reverse causality. Therefore, the aforementioned analyses were performed by excluding patients who were diagnosed with pneumonia (during either inpatient or outpatient visits) in the hazard period. It should be noted that the primary analysis was based on the results after excluding pneumonia during the hazard period in the study. (Page 8, Lines 164-167)

Regarding other infectious diseases, such as infective endocarditis (IE), we recognize its significance as a major cause of valvular regurgitation. However, IE cases (n = 188) were explicitly excluded from our entire study period during the initial screening process, as detailed in our flowchart (Fig 1). This exclusion ensures that the effects we observed are more accurately attributed to fluoroquinolone use rather than being confounded by IE.

3. The authors also observed the association between other antibiotics and valuvlar insufficency. What is the plausibility of this observation?

Response: Thank you for your comment. In our study, we observed an association between most antibiotics and valvular heart disease. This observation may be attributed to the treatment of certain infections with symptoms similar to valvular insufficiency. Clinically, it can be challenging to differentiate between the symptoms of pneumonia and valvular insufficiency before echocardiographic evaluation. Both conditions share common signs and symptoms, such as exertional breathlessness, nocturnal cough, and paroxysmal nocturnal dyspnea.

Before excluding pneumonia diagnosed within 2 months before the index date, our results indicated an association between fluoroquinolones (FQs) and aortic regurgitation (AR) or mitral regurgitation (MR), as well as any AR and any MR. Additionally, we observed similar associations between other antibiotics and valvular insufficiency. To strengthen the robustness of our findings, we excluded individuals diagnosed with pneumonia within 2 months before the index date. After this exclusion, only FQs remained significantly associated with a greater risk of AR or MR and any MR. We have revised the STATISTICAL ANALYSIS subsection in the METHODS section to emphasize that the results after excluding pneumonia during the hazard period were considered the primary analyses. (Page 8, Lines 164-167)

As shown in Table 3, FQs exhibited a significant and higher risk ratio compared to other antibiotics, further emphasizing the importance of our focus on discussing the results related to FQs.

 

Reviewer #2: This was a very interesting study that explored the relationship between FQs and the onset of valvular heart disease. However, I have some questions about this paper.

1.Rather than suggesting that exposure to FQs poses a risk for MR and AR, the results presented in this paper give the impression that antibiotics are associated with the development of valvular heart disease.

Response: Thank you for your comment. In our study, we observed an association between most antibiotics and valvular heart disease. This observation may be attributed to the treatment of certain infections with symptoms similar to valvular insufficiency. Clinically, it can be challenging to differentiate between the symptoms of pneumonia and valvular insufficiency before echocardiographic evaluation. Both conditions share common signs and symptoms, such as exertional breathlessness, nocturnal cough, and paroxysmal nocturnal dyspnea.

Before excluding pneumonia diagnosed within 2 months before the index date, our results indicated an association between fluoroquinolones (FQs) and aortic regurgitation (AR) or mitral regurgitation (MR), as well as any AR and any MR. Additionally, we observed similar associations between other antibiotics and valvular insufficiency. To strengthen the robustness of our findings, we excluded individuals diagnosed with pneumonia within 2 months before the index date. Even after this exclusion, only FQs remained significantly associated with a greater risk of AR or MR and any MR. We have revised the STATISTICAL ANALYSIS subsection in the METHODS section to emphasize that the results after excluding pneumonia during the hazard period were considered the primary analyses. (Page 8, Lines 164-167)

As shown in Table 3, FQs exhibited a significant and higher risk ratio compared to other antibiotics, further emphasizing the importance of our focus on discussing the results related to FQs.

2.Although the purpose of thi study was to explore the risks of FQs, it is not clear why amoxicillin, TMP/SMX, AMC and ASB, and second-generation cephalosporin drugs were selected in addition to FQs.

Response: Thank you for your insightful question. These antibiotics (amoxicillin, TMP/SMX, AMC, ASB, and second-generation cephalosporins) were selected as negative control exposures to ensure the robustness of our results. By comparing fluoroquinolones to these commonly used antibiotics, we aimed to control for confounding factors and strengthen the validity of our findings related to the cardiovascular risks associated with fluoroquinolone use. These antibiotics were chosen because their indications are generally similar to those of fluoroquinolones, according to treatment guidelines in Taiwan and other regions of Asia. We have supplemented this information in the STUDY DESIGN subsection of the METHODS section in the revised manuscript. (Page 6, Lines 130-136)

3. A national cohort study in Taiwan did not show any association between FQs and the onset of valvular heart disease. Please comment on the differences with this previous study.

Clin Pharmacol Ther. 2024 Jan;115(1):147-157. doi: 10.1002/cpt.3084. Epub 2023 Nov 13.

https://ascpt.onlinelibrary.wiley.com/doi/10.1002/cpt.3084

Response: Thank you for bringing this recent study to our attention. We have included the discussion of this study in our manuscript. Indeed, the findings from our study indicating an increased risk of aortic and mitral valve regurgitation in patients treated with fluoroquinolones (FQs) contrast with the results of the recent nationwide cohort study in Taiwan, which did not show a significant association between FQs and higher rates of valvular regurgitation. Clin Pharmacol Ther. (2024 Jan;115(1):147-157. doi: 10.1002/cpt.3084. Epub 2023 Nov 13) 

Several key methodological differences may account for these contrasting findings. Our study utilized a unidirectional case-crossover design, comparing hazard periods and referent periods within the same individuals. This design helps to minimize within-person time-invariant confounding factors such as genetic predispositions, thereby reducing potential biases associated with external control groups. Conversely, the previous study's active comparative design, comparing FQs with amoxicillin/clavulanate among different individuals, can introduce various confounding factors and biases that are not present in a case-crossover design. Additionally, their study differentiated between respiratory and non-respiratory FQs, which could further complicate the interpretation of their results. Despite these conflicting results, it is crucial to recognize the inherent limitations and differences in study designs when interpreting these findings. Further research is warranted to confirm or refute the association between FQs and the risk of aortic and mitral regurgitation, taking into account the various confounding factors and biases.

(Page 3, Line 81; Page 4, Lines 82-83; Page 14, Line 272; Page 15, Lines 273-289)

(Reference 12)

---

## [Decision Letter · Decision Letter 1]

8 Jul 2024

Use of fluoroquinolones and the risk of aortic and mitral regurgitation: a nationwide case-crossover study

PONE-D-24-09908R1

Dear Dr. Chen,

We’re pleased to inform you that your manuscript has been judged scientifically suitable for publication and will be formally accepted for publication once it meets all outstanding technical requirements.

Kind regards,

Yoshihiro Fukumoto

Academic Editor

PLOS ONE

Additional Editor Comments (optional):

Reviewers' comments:

Reviewer's Responses to Questions

**Comments to the Author**

1. If the authors have adequately addressed your comments raised in a previous round of review and you feel that this manuscript is now acceptable for publication, you may indicate that here to bypass the “Comments to the Author” section, enter your conflict of interest statement in the “Confidential to Editor” section, and submit your "Accept" recommendation.

Reviewer #1: All comments have been addressed

Reviewer #2: All comments have been addressed

2. Is the manuscript technically sound, and do the data support the conclusions?

Reviewer #1: Yes

Reviewer #2: Yes

3. Has the statistical analysis been performed appropriately and rigorously? 

Reviewer #1: Yes

Reviewer #2: Yes

4. Have the authors made all data underlying the findings in their manuscript fully available?

Reviewer #1: Yes

Reviewer #2: Yes

5. Is the manuscript presented in an intelligible fashion and written in standard English?

Reviewer #1: Yes

Reviewer #2: Yes

6. Review Comments to the Author

Reviewer #1: (No Response)

Reviewer #2: I think that the authors have adequately addressed the comments made by the reviewers in the revised version of the manuscript. Therefore, I have no further comments.

7. PLOS authors have the option to publish the peer review history of their article (what does this mean?). If published, this will include your full peer review and any attached files.

Reviewer #1: No

Reviewer #2: No

---

## [Editor Report · Acceptance letter]

15 Jul 2024

PONE-D-24-09908R1 

PLOS ONE

Dear Dr. Chen, 

I'm pleased to inform you that your manuscript has been deemed suitable for publication in PLOS ONE. Congratulations! Your manuscript is now being handed over to our production team.

Kind regards, 

on behalf of

Dr. Yoshihiro Fukumoto 

Academic Editor

PLOS ONE